# Precipitation Microphysics during the Extreme Meiyu Period in 2020

**Aoqi Zhang** [1], **Yilun Chen** [1,*], **Shengnan Zhou** [2], **Shumin Chen** [1] and **Weibiao Li** [1]

[1] Southern Marine Science and Engineering Guangdong Laboratory (Zhuhai), School of Atmospheric Sciences, Sun Yat-sen University, Zhuhai 519082, China; zhangaoq3@mail.sysu.edu.cn (A.Z.); chenshm35@mail.sysu.edu.cn (S.C.); esslwb@mail.sysu.edu.cn (W.L.)

[2] Anhui Meteorological Observatory, Hefei 230031, China; zhoushengnan410557@cma.cn

[*] Correspondence: chenylun3@mail.sysu.edu.cn; Tel.: +86-189-4989-1911

**Abstract:** Previous studies have reported the large-scale meteorological conditions and dynamic causes of the extreme period of meiyu rainfall in 2020. However, the microphysical properties of meiyu precipitation during this period remain unclear. We used the Global Precipitation Measurement 2ADPR orbital precipitation dataset, the IMERG gridded precipitation dataset and the ERA5 reanalysis dataset to study the characteristics of meiyu precipitation over the Yangtze Plain during the extreme meiyu period in 2020 and historical meiyu periods from 2014 to 2019. The results showed that the average daily rainfall during the 2020 meiyu period was 1.5 times higher than the historical average as a result of the super-strong water vapor flux in the low- to mid-level layers of the atmosphere. The amplitude of nocturnal low-level water vapor transport during the 2020 meiyu period was twice the historical average and, therefore, the diurnal peak of meiyu rainfall at 0630 LST in 2020 was significantly earlier than the historical average. The moisture transport was the dominant moisture supply for precipitation during the 2020 meiyu period, whereas the moisture convection contributed less than in the meiyu periods of 2014–2019. This led to the precipitation in the 2020 meiyu period having a higher concentration of smaller droplets than the historical average. There were lower proportions of size-sorting evaporation and break-up processes in the liquid-phase precipitation processes in the 2020 meiyu than the historical average, but a higher proportion of coalescence processes. These results provide a factual basis for the simulation and forecast of precipitation during extreme meiyu periods.

**Keywords:** precipitation microphysics; moisture transport; GPM DPR; diurnal variation of precipitation; IMERG

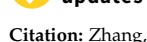



## 1. Introduction

The meiyu period, also called the baiu or changma period, refers to a continuous period of rainy days that occurs every year from the Yangtze Plain to southern Japan between mid-June and mid-July [1–3]. Meiyu precipitation occurs in the frontal zone on the edge of the western Pacific subtropical high, commonly known as the meiyu front, which is a quasi-stationary front system formed by the interaction of polar and subtropical air masses [4,5]. Affected by many factors—including the western Pacific subtropical high, the El Niño–Southern Oscillation and the condition of the Indian Ocean—the onset and end dates of the meiyu period vary between years and in different regions [6–8].

Record-breaking meiyu rainfall, the highest since 1951, was recorded over the Yangtze basin, Korea and Japan during 2020 [9]. Despite slight differences in various reports [10–12], the meiyu period over the Yangtze Plain in 2020 started on about 9 June and ended by 31 July. As a result of such a long meiyu period, the total amount of rainfall over the Yangtze Plain reached 700 mm during June–July 2020, twice the climatological average of 1981–2010 [13]. This concentrated precipitation during the extreme meiyu period in

2020 caused serious economic losses and casualties and has therefore attracted widespread attention from the academic community and society [14].

There has been much research into the causes and predictability of the 2020 extreme meiyu period. Using a Lagragian model, Zhang et al. (2021) reported that the moisture content during the 2020 meiyu period was 1.5–1.6 times the climatological mean [12]. The moisture content was determined by the dynamic circulation and contributed directly to the dramatic meiyu precipitation over the Yangtze Valley [10]. Previous studies have shown that the main dynamic force in meiyu rainfall is a strong El Niño–Southern Oscillation, which leads to an anomalous anticyclone in the Northwestern Pacific and the enhanced transport of moisture content and increased rainfall [15]. However, the El Niño–Southern Oscillation was weak in the 2019–2020 winter and, therefore, another mechanism was suggested in which extreme conditions in the Indian Ocean in the boreal autumn of 2019 were instrumental in triggering the record-breaking meiyu in the summer of 2020 [11,16–19].

The extreme meiyu period in 2020 was also related to the low Arctic ice cover in late spring to early summer 2020 [20], anomalous high sea surface pressure over the South China Sea [21], the Silk Road wave train [22], the Madden–Julian Oscillation [13] and the North Atlantic Oscillation [23]. In terms of predictability, current models predicted rainfall exceeding the historical average, but underestimated about 80% of the total meiyu rainfall in 2020, which may be linked with the poor prediction of the change in the intensity of the westerly jet, the Silk Road pattern, the Asian two-ridge–one-trough pattern and other factors [24–27].

Significant interannual variations were reported in the duration and total amount of rainfall in the meiyu period [8,28,29]. Volonté et al. (2021) used data from IMERG to investigate the magnitude and scale of meiyu rain bands over China and showed that the average precipitation and number of days with extreme precipitation during the 2020 meiyu period were the highest since 2000 [30]. Apart from the amount of rainfall, other properties of the meiyu precipitation also show interannual variations. For instance, Cui et al. (2020) investigated the properties of mesoscale convective systems during the meiyu period over China and found large interannual variations in the number, size and lifetime of clouds and the intensity of rainfall [31]. Zhang et al. (2020) reported two diurnal peaks of rainfall over the Yangtze Plain in the early morning and afternoon during atypical meiyu years, such as 2019, whereas there was only one morning peak during normal meiyu years, such as 2016 [32]. These interannual variations in meiyu rainfall are strongly related to large-scale environments, especially the intensity of the westerly nocturnal low-level jet [33].

By contrast, there has been less research on the interannual variations in the vertical structure and microphysics of meiyu precipitation. Most research has focused on statistical averages or case studies of meiyu microphysics. For instance, Wang and Yang (2003) used ground-based radar observations and model simulations to investigate the microphysical processes of one meiyu precipitating cloud system and suggested that mixed-phase cloud processes have the most important role in the triggering of heavy convective meiyu rainfall [34]. Using Tropical Rainfall Measuring Mission (TRMM) Precipitation Radar (PR) observations, Yokoyama et al. (2014) showed that the characteristics of meiyu rainfall are obviously different between the southern and northern sides of the meiyu–baiu front [35]. Chen et al. (2019) used ground-based polarimetric radar and disdrometer measurements in the analysis of a case of meiyu precipitation and found that meiyu precipitation occurs in wet, stable environments, thus active warm rain processes such as coalescence occur within the precipitation [36].

Wen et al. (2020) used ground-based polarimetric radar detections to study the microphysical features of meiyu precipitation using statistics [37]. They reported that both convective and stratiform precipitations during the meiyu period are dominated by small droplets, and the increase in precipitation intensity is mostly attributed to the higher concentration of smaller droplets. Based on Global Precipitation Mission (GPM) Dual-frequency Precipitation Radar (DPR) measurements, Sun et al. (2020) investigated the vertical structures of three typical meiyu rainfall events at different life stages and found

that the precipitation microphysics were different during the different life stages of the precipitating systems [38]. An in-depth analysis of the microphysical processes within meiyu precipitation over a period of extreme meiyu rainfall and its variations from normal meiyu years will provide a factual basis for the simulation and forecast of meiyu rainfall in these extreme periods.

We combined three-dimensional measurements and droplet size distribution (DSD) information from the GPM 2ADPR orbital product, the IMERG half-hourly gridded precipitation data and the ERA5 hourly reanalysis dataset to investigate precipitation microphysics over the Yangtze Plain during the extreme meiyu period in 2020.

## 2. Data and Methods

### 2.1. Datasets

As a successor to the TRMM PR, the GPM DPR covers the globe from 67° S to 67° N and has provided joint observations of the three-dimensional structure of precipitation from both Ku-band (KuPR) and Ka-band radar (KaPR) since February 2014. The KuPR is similar to the TRMM PR, which works in a normal scan mode with 49 pixels per track. The KaPR works in both matched scan and high-sensitivity scan modes. Only the KaPR matched scan pixels were matched to the central 25 pixels of KuPR prior to May 2018, whereas all 49 KaPR pixels were matched to KuPR after May 2018 [39]. Based on the matched pixels of the KuPR and KaPR, the GPM precipitation processing system develops a series of dual-frequency algorithms and provides users with the GPM 2ADPR dual-frequency precipitation product [40]. We used the Ku-band corrected reflectivity profile, the DSD profile, the near-surface rain rate and other precipitation information from the GPM 2ADPR product with a horizontal distribution of about 5 km and a vertical interval of 125 m.

We also used the Integrated Multi-satellite Retrievals for the GPM (IMERG) precipitation dataset to investigate the horizontal distribution and diurnal cycle of meiyu rainfall. IMERG combines precipitation estimates from satellite visible–infrared sensors and microwave together with observations from rain gauges. IMERG has a horizontal resolution of (0.1° × 0.1°) and a temporal resolution of 0.5 h. As a result of the calculation time and usage data, IMERG is subdivided into early run, late run and final run products. We used the IMERG final run version 6 product. The quality index of the IMERG product was high over the Yangtze Plain as a result of the high number of rain gauge and microwave-based observations [41].

We used the wind, specific humidity, geopotential and other environmental information from the ERA5 dataset at 500 and 850 hPa from the ERA5 reanalysis dataset provided by the European Centre for Medium-Range Weather Forecasts [42]. The horizontal resolution of the dataset is (0.25° × 0.25°) at a temporal interval of 1 h.

### 2.2. Methods

Consistent with Zhang et al. (2020) [32], our study region covered the range of the Yangtze Plain (28–34° N, 110–122° E). Since the GPM 2ADPR dataset starts from 2014, we focused on meiyu precipitation from 2014 to 2020. The onset and end dates of the meiyu periods over the Yangtze Plain were provided by the Anhui Meteorological Observatory (Table 1). Two main criteria were employed including continuously cloudy or rainy days and sustained meiyu fronts. During 2020, the meiyu period lasted for about 53 days, which is about 2.5 times the average of the meiyu periods from 2014 to 2019 (about 26.7 days).

**Table 1.** Onset date, end date and duration of meiyu periods during the time period 2014–2020 over the Yangtze Plain provided by the Anhui Meteorological Observatory.

| Year | Onset Date | End Date | Duration (Days) |
|------|------------|----------|-----------------|
| 2014 | 25 June | 5 July | 11 |
| 2015 | 24 June | 24 July | 31 |
| 2016 | 20 June | 21 July | 32 |
| 2017 | 30 June | 10 July | 11 |
| 2018 | 19 June | 11 July | 23 |
| 2019 | 17 June | 8 July | 22 |
| 2020 | 9 June | 31 July | 53 |

The horizontal water vapor flux ($F_H$; units: g m$^{-1}$ s$^{-1}$ Pa$^{-1}$) at 500 or 850 hPa was calculated on each ERA5 grid with a time step of 1 h using the Equation (1):

$$F_H = \frac{vq}{g} \qquad (1)$$

where $v$ (units: m s$^{-1}$) and $q$ (units: g kg$^{-1}$) indicate the horizontal wind and specific humidity at 500 or 850 hPa and $g$ is the acceleration due to gravity (9.80665 m s$^{-2}$). We also calculated the geostrophic wind ($v_g$; units: m s$^{-1}$) using the geopotential height ($z$; units: m) from the ERA5 dataset:

$$v_g = -\frac{g}{f}\frac{\partial z}{\partial N} \qquad (2)$$

$$f = 2\omega \sin \varphi \qquad (3)$$

where $f$ is the geostrophic parameter (units: $s^{-1}$), $N$ is the direction of the geopotential gradient (including x and y direction), $\omega$ (7.292 × 10$^{-5}$ *rad s*$^{-1}$) is the rotational angular velocity of the Earth and $\varphi$ is the latitude. The ageostrophic winds ($v_a$; units: m s$^{-1}$) were then calculated by:

$$v_a = v - v_g \qquad (4)$$

## 3. Results

### 3.1. Horizontal Distribution and Diurnal Variation

Many previous studies have simply regarded precipitation events in June–July as meiyu precipitation [30,35]. However, meiyu precipitation differs from other precipitation in the boreal summer [43,44]. Specifically, meiyu precipitation usually occurs in the morning and is dominated by stratiform precipitation, while other precipitation in the boreal summer occurs more frequently in the afternoon with active convections. Therefore, we only selected precipitation events that occurred during the exact meiyu period.

Figure 1 presents the horizontal distributions of the average daily rainfall over the extreme meiyu period in 2020 and the climatology from 2014 to 2019 using the IMERG precipitation dataset. During the extreme meiyu period in 2020 (Figure 1a), precipitation in eastern China was concentrated in the meiyu belt along the Yangtze River, with an average daily rainfall in most areas south or north of the Yangtze Plain <5 mm day$^{-1}$. The average daily rainfall over the Yangtze Plain was between 15 and 21 mm day$^{-1}$, whereas the highest average daily rainfall near Dabie Mountain exceeded 25 mm day$^{-1}$ due to topographic uplift [45]. There was more precipitation over the southeast China during the meiyu periods in 2014–2019 (Figure 1b), with an average daily rainfall of about 7–9 mm day$^{-1}$. The average daily rainfall over the Yangtze Plain was between 9 and 13 mm day$^{-1}$, with a peak value of about 18 mm day$^{-1}$.

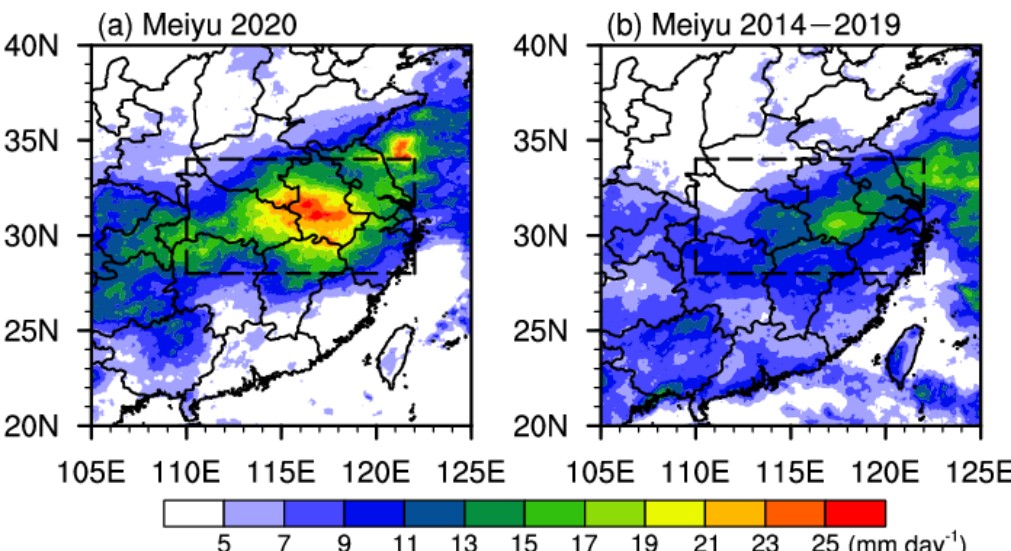

**Figure 1.** Horizontal distribution of the average daily rainfall during meiyu periods in (**a**) 2020 and (**b**) 2014–2019, derived from IMERG precipitation dataset. The dashed rectangle indicates the range of the study region.

During the extreme meiyu period in 2020, the average daily rainfall over the Yangtze Plain reached about 1.5 times the historical average (Figure 1), whereas the duration reached 2 times the historical average (Table 1). The total amount of rainfall during the extreme meiyu period in 2020 therefore reached three times the climatological mean and caused severe flooding disasters.

Meiyu precipitation is strongly linked with the westerly transport of moisture and therefore the location of the western Pacific subtropical high [2]. Figures 2 and 3 show the average horizontal distribution of ERA5 geopotential height and water vapor flux at 500 and 850 hPa for the meiyu periods in 2020 and 2014–2019. Specifically, since the specific humidity at 500 hPa is small, the water vapor flux on 500 hPa is dominated by the wind field. During boreal summer, the western Pacific subtropical high can usually be represented by the area with a 500 hPa geopotential height higher than 5880 m [46]. For the meiyu periods in 2014–2019, the average subtropical high was located to the east of Taiwan island; the ridge of the subtropical high was located at about 24° N with the western ridge point at 122° E (Figure 2b). For the extreme meiyu period in 2020, the average subtropical high covered a broader area, with the ridge of the subtropical high at about 20° N and the western ridge point at 110° E (Figure 2a). This meant that southeast China was covered by the subtropical high, and precipitation was suppressed in this region (Figure 1a). The contours of the 500 hPa geopotential height over the Yangtze Plain were much denser in the 2020 meiyu period than in the 2014–2019 meiyu period (Figure 2). The 500 hPa wind and water vapor transport during the meiyu period in 2020 were therefore stronger than the historical average.

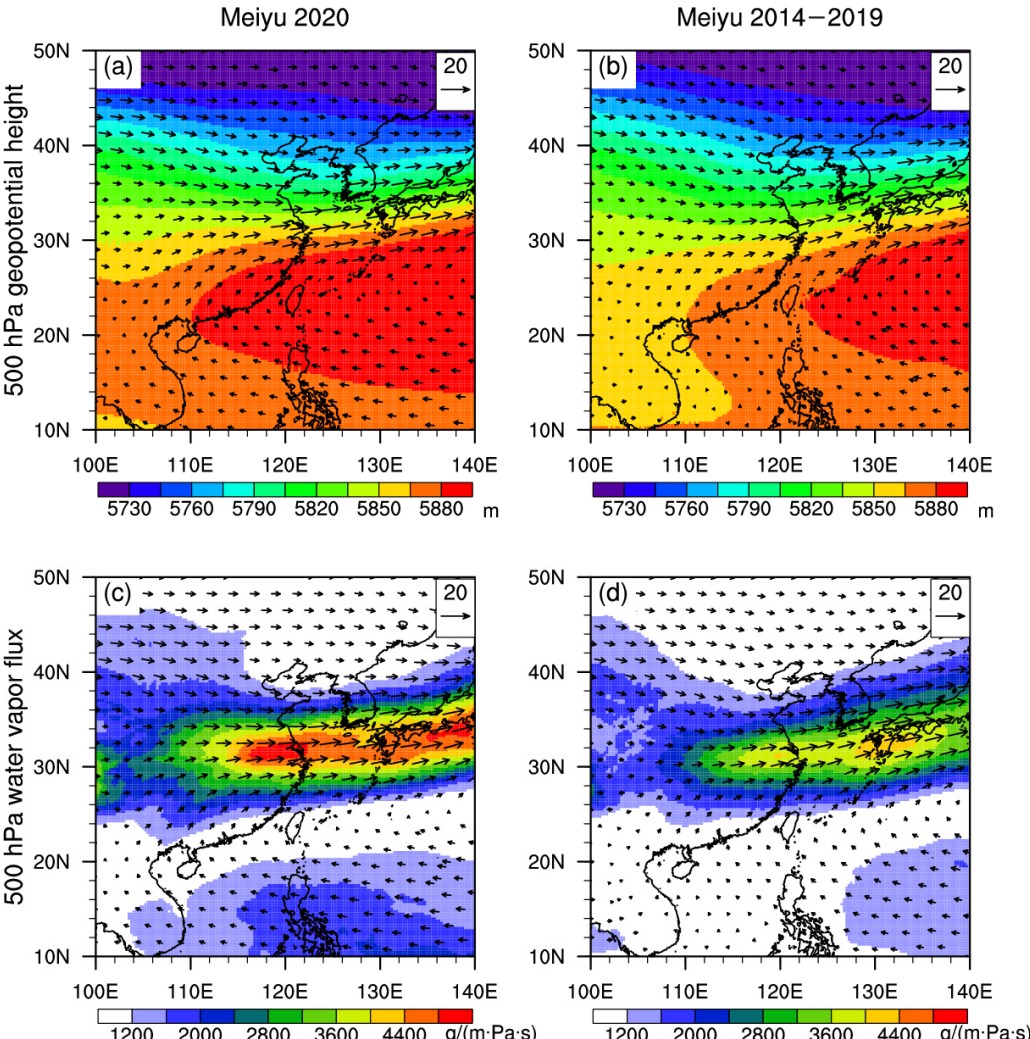

**Figure 2.** Average horizontal distribution of the (**a**,**b**) 500 hPa geopotential height and (**c**,**d**) water vapor flux overlapped with the winds during the meiyu periods in 2020 (left-hand panels) and 2014–2019 (right-hand panels), derived from the hourly data in the ERA5 dataset.

The low-level jet is crucial in triggering heavy precipitation and is often indicated by the horizontal wind or water vapor flux at 925 hPa [47]. However, considering that the terrains to the south and west of the study area may be higher than the 925 hPa geopotential height, we focused on the 850 hPa water vapor flux. During the extreme meiyu period in 2020, the low-level southwesterly wind over the Yangtze Plain and southeast China was affected by the strong anticyclone in the western Pacific and was significantly stronger than that in 2014–2019 (Figure 3a,b). Affected by the wind field, the strongest 850 hPa water vapor flux reached 16,000 g m$^{-1}$ Pa$^{-1}$ s$^{-1}$, which is 1.6 times the historical record (about 10,000 g m$^{-1}$ Pa$^{-1}$ s$^{-1}$). The super-strong water vapor flux decayed rapidly over the Yangtze Plain, leading to numerous precipitation events.

The diurnal variation in precipitation in boreal summer over the Yangtze Plain has received widespread attention. Yu et al. (2007) revealed that precipitation in boreal summer over the Yangtze Plain presents two rainfall peaks in the morning and late afternoon [48]. Although meiyu precipitation also happens in boreal summer, the diurnal variation in meiyu precipitation is different from the total precipitation in boreal summer and varies between years. In normal meiyu years, meiyu precipitation usually presents a single morning peak, which is linked with the nocturnal low-level jet [49,50]. By contrast, the meiyu precipitation in atypical meiyu years shows two diurnal peaks in the early morning and afternoon [32].

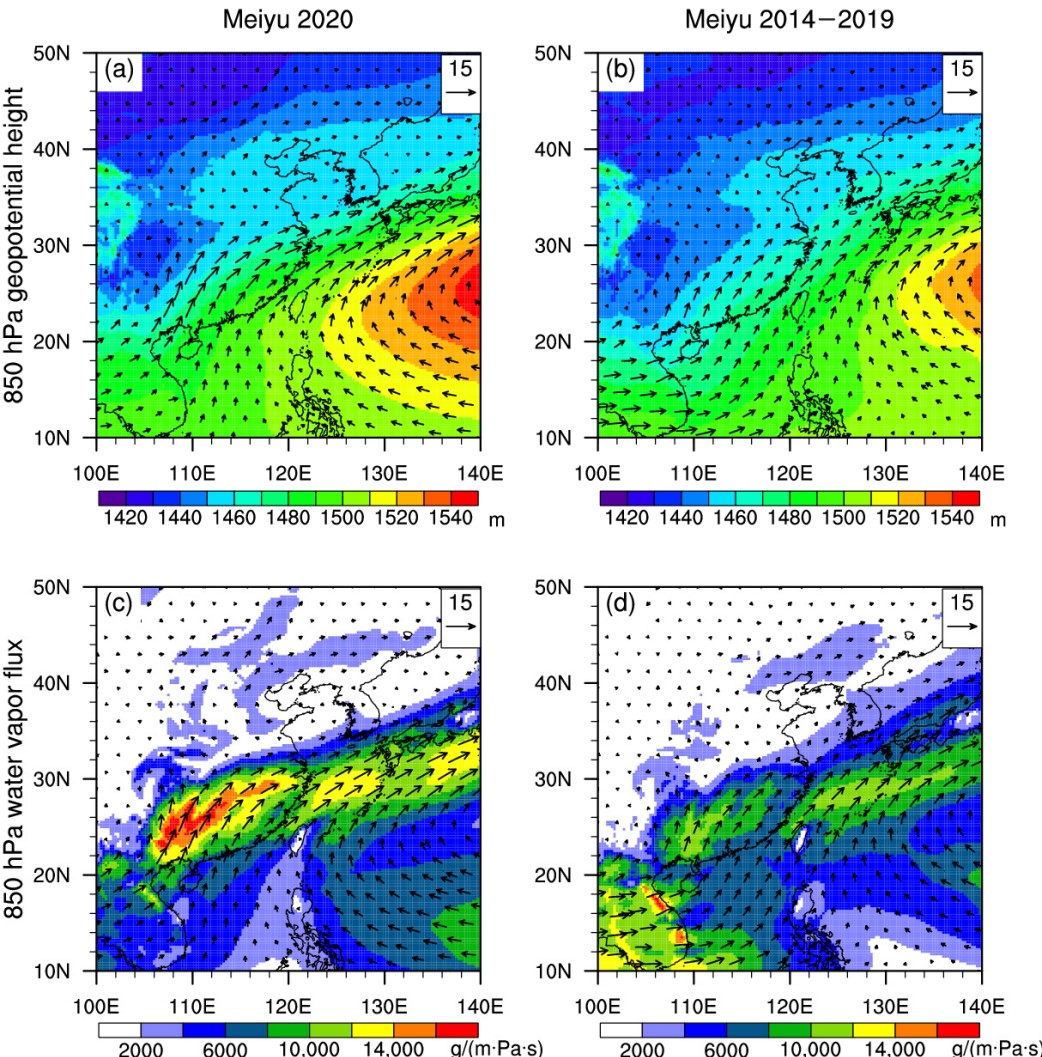

**Figure 3.** Average horizontal distribution of the (**a**,**b**) 850 hPa geopotential height and (**c**,**d**) water vapor flux overlapped with the winds during the meiyu periods in 2020 (left-hand panels) and 2014–2019 (right-hand panels), derived from the hourly data in the ERA5 dataset.

Based on these results, we further investigated the diurnal meiyu precipitation during the extreme meiyu period in 2020. Figure 4 shows that during the meiyu period in 2020, the average rainfall over the Yangtze Plain at each local time exceeded 0.5 mm h$^{-1}$, which was 0.15–0.3 mm h$^{-1}$ higher than the historical average of 2014–2019. Despite the small fluctuations caused by the sample size, the diurnal rainfall peak of meiyu precipitation in 2020 appeared at around 0630 LST in the morning, with an average rainfall of 0.75 mm h$^{-1}$, which is earlier than in normal meiyu years with a rainfall peak at about 0930 LST [32,50]. For comparison, the diurnal variation in meiyu precipitation in 2014–2019 presents a trapezoidal distribution with a peak time from 0930 to 1430 LST, with an average rainfall of about 0.44 mm h$^{-1}$.

To understand why the diurnal rainfall peak of meiyu precipitation in 2020 was higher than that in 2014–2019, we determined the diurnal variations in the 850 hPa water vapor flux using the total wind and ageostrophic wind. The total wind is composited of the ageostrophic and geostrophic winds. The ageostrophic wind includes the thermogenic wind and the antitriptic wind (usually small), so it is mainly controlled by the thermal differences between regions. As a result, the ageostrophic wind over the Yangtze Plain is similar in the meiyu periods of different years, but has significant diurnal variations. By contrast, the geostrophic wind is completely controlled by the pressure gradient, so it varies

significantly between years but has almost no diurnal variation. The diurnal variation of the total water vapor flux is therefore controlled by the ageostrophic wind, whereas the interannual variation is controlled by the geostrophic wind.

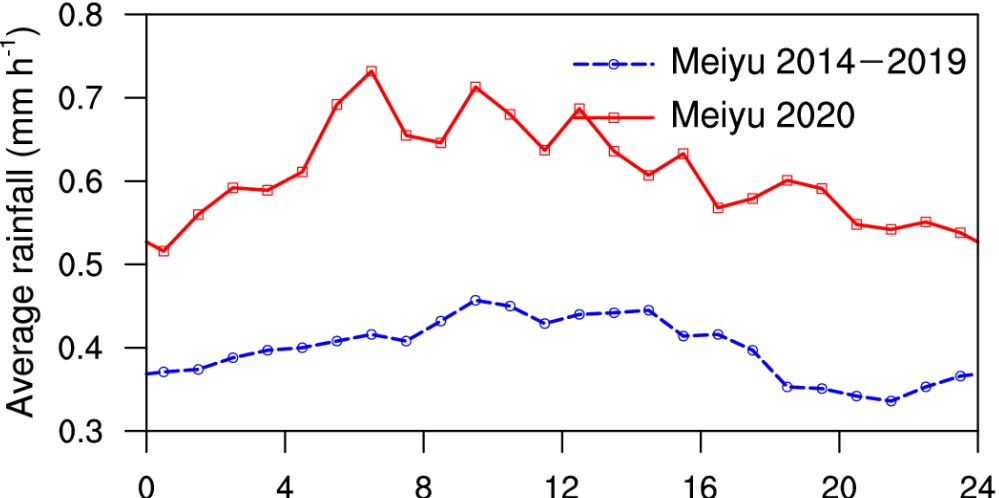

**Figure 4.** Diurnal variation in meiyu precipitation during the meiyu periods in 2020 and 2014–2019 derived from the IMERG dataset.

At 0200–0800 LST, the ageostrophic water vapor flux was southwesterly (Figure 5), similar to the direction of the geostrophic wind. Under this condition, the diurnal peak of the total water vapor flux over the Yangtze Plain occurred at about 0400 LST for the meiyu periods in both 2020 and 2014–2019. During the extreme meiyu period in 2020, the geostrophic wind was significantly stronger than the historical average, so the total 850 hPa water vapor flux was twice that in 2014–2019. The super-strong nocturnal low-level moisture transport (the low-level jet) promoted rainfall in the morning, resulting in an earlier diurnal peak of meiyu rainfall in 2020 (Figure 4).

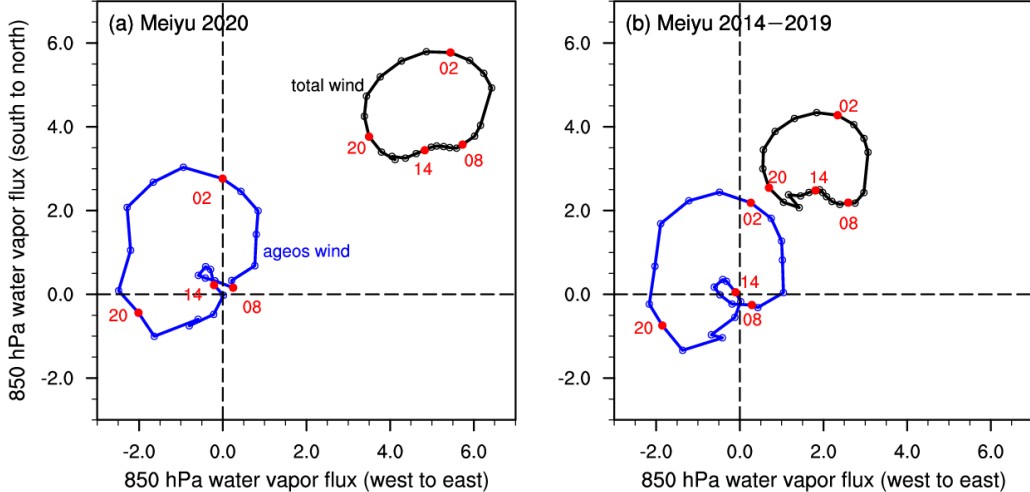

**Figure 5.** Diurnal variation in 850 hPa water vapor flux (units: kg m$^{-1}$ Pa$^{-1}$ s$^{-1}$) during the meiyu periods in 2020 and 2014–2019 derived from the ERA5 dataset. The red markers and labels 02, 08, 14 and 20 indicate different local times. The label ageos wind indicates water vapor flux by ageostrophic wind.

In addition, our analysis shows that due to the cloud's shielding of solar heating, the maximum temperature within a day was, on average, lower during meiyu days in 2020 than on those in 2014–2019 (not shown), which would suppress the convective activities in the afternoon.

*3.2. Vertical Structure and Microphysics*

Previous studies based on radar observations have laid a good foundation for understanding the microphysics of meiyu precipitation [37,51]. However, the vertical structure and microphysics of meiyu precipitation during the extreme meiyu period in 2020 are not yet known. We therefore used the GPM DPR orbital product 2ADPR to investigate this issue. Considering that the moisture and thermal conditions are significantly different between morning and afternoon, meiyu rainfalls were classified into morning rainfall (00–12 LST) and afternoon rainfall (12–24 LST) for statistics.

The basic information for meiyu precipitation in 2020 and 2014–2019 was provided in Table 2. Due to the long duration of the meiyu period and frequent meiyu precipitation in 2020, the detected sample size over the Yangtze Plain in 2020 (71,325) reached almost half the sample size from 2014 to 2019 (143,569). Specifically, at 00–12 LST, the ratio of convective precipitation for meiyu precipitation in 2020 (8.47%) was close to that in 2014–2019 (7.81%), with similar storm-top height (6.8 km for both 2020 and 2014–2019). At 12–24 LST, the ratio of convective precipitation reached 11.83% for meiyu precipitation in 2014–2019, which was obviously higher than that in 2020 (8.11%); the average storm-top height for convective precipitation in 2014–2019 (8.7 km) was also higher than that in 2020 (7.82 km). Therefore, it is concluded that the convection activities had stronger contributions for afternoon precipitations in 2014–2019 than in 2020.

**Table 2.** The ratio, average near-surface rain rate (*RR*) and average storm-top height (*STH*) for precipitations of different rain type over the Yangtze Plain during the extreme meiyu period in 2020 and the meiyu periods from 2014 to 2019, derived from GPM 2ADPR dataset. Rain types include stratiform, convective, shallow and other (less than 1%, not shown).

| | | 00–12 LST | | 12–24 LST | |
|---|---|---|---|---|---|
| | | 2014–2019 | 2020 | 2014–2019 | 2020 |
| | Total Sample Size | 70,737 | 29,444 | 72,832 | 41,881 |
| | ratio | 77.06% | 75.35% | 73.12% | 74.34% |
| Stratiform | $RR$ (mm h$^{-1}$) | 2.37 | 2.51 | 1.76 | 1.91 |
| | $STH$ (km) | 6.80 | 6.80 | 6.78 | 6.84 |
| | ratio | 7.81% | 8.47% | 11.83% | 8.11% |
| Convective | $RR$ (mm h$^{-1}$) | 10.63 | 9.75 | 8.38 | 7.92 |
| | $STH$ (km) | 7.67 | 7.61 | 8.70 | 7.82 |
| | ratio | 14.18% | 15.42% | 13.97% | 16.84% |
| Shallow | $RR$ (mm h$^{-1}$) | 1.20 | 1.29 | 1.26 | 1.28 |
| | $STH$ (km) | 3.20 | 3.22 | 3.26 | 3.30 |

Figure 6 shows the probability distribution functions (PDFs) of the near-surface rainfall contribution (rain rate × number) and storm-top height during the meiyu periods in 2020 and 2014–2019. We used the PDF of near-surface rainfall contribution rather than near-surface rain rate because it can better reflect the contribution of rain rate to the total amount of rainfall [52].

At 00–12 LST, when the atmospheric thermal conditions were quite stable, meiyu precipitation mostly appeared as stratiform or shallow rainfall in both 2020 and 2014–2019 (Table 2). The PDFs of the storm-top height were quite similar between meiyu rainfall in 2020 and 2014–2019, with peaks at about 5.6 km with a proportion of 6.8% (Figure 6b). Due to the stronger water vapor flux in the low- to mid-level layers of atmosphere, the precipitations with rain rate from 2 to 25 mm h$^{-1}$ had a larger contribution to the total amount of rainfall in the 2020 meiyu period than in the 2014–2019 meiyu periods (Figure 6a).

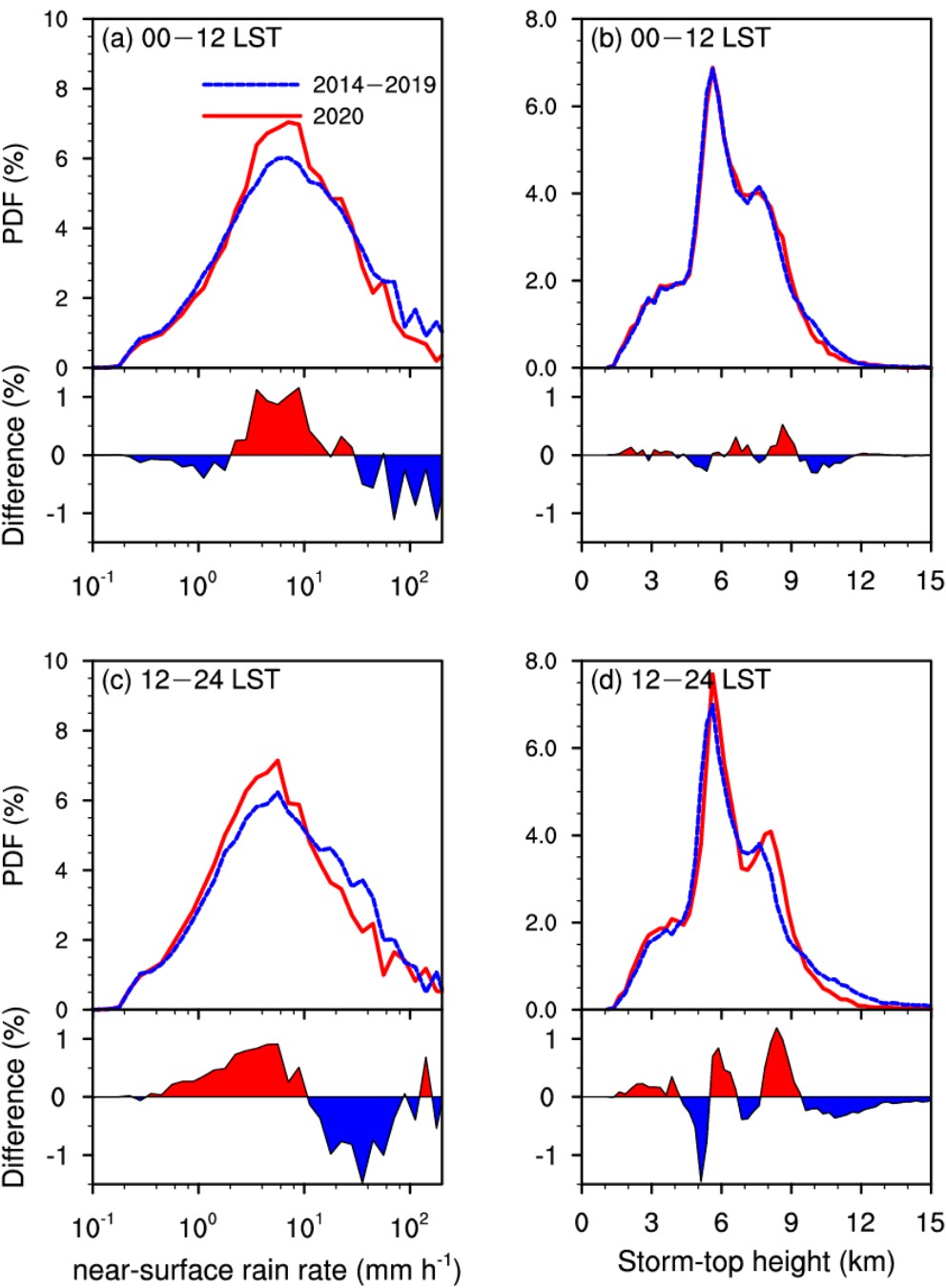

**Figure 6.** PDFs of the (**a**,**c**) near-surface rainfall contribution (rain rate × number) and (**b**,**d**) storm-top height of meiyu precipitation during the extreme meiyu period in 2020 and the meiyu periods from 2014 to 2019 and their differences. The top panels are from precipitation events that occurred in the morning (00–12 LST), and the bottom panels are from precipitation events that occurred in the afternoon (12–24 LST). The spacing of the near-surface rain rate is constant in the log coordinate ($\Delta(lgRR) = 0.1$), whereas the spacing of the storm-top height is 0.25 km.

At 12–24 LST, both the PDFs of the near-surface rainfall contribution in the 2020 and 2014–2019 meiyu periods peaked at about 6 mm h$^{-1}$ with a proportion of about 6–7% (Figure 6c). However, during the extreme meiyu period in 2020, weak precipitation with a rain rate < 10 mm h$^{-1}$ contributed more to the total amount of rainfall than in 2014–2019, whereas it was just the opposite for precipitation with a rain rate > 10 mm h$^{-1}$ (Figure 6c). The average storm-top height for afternoon meiyu rainfall was lower in 2020 than 2014–2019,

with less proportions of storm-top height > 10 km (Figure 6d). We suggest it was due to the frequent convective activities in the afternoon precipitation in 2014–2019 meiyu periods (Table 2), thus more short-term heavy precipitation occurred in 2014–2019 than in 2020.

Contoured frequency by altitude diagrams (CFADs) of the radar reflectivity are widely used to analyze the vertical structure of clouds or precipitation [53]. The CFAD can be normalized in different ways, including the overall maximum, the overall sum and the sums of each height level. We used the overall sum to normalize the CFAD—that is, the sum of all the grids in the CFAD equals 1. Specifically, the interval of each grid is 0.5 km (height) × 1 dBZ (reflectivity).

Figure 7 shows the CFADs of the Ku-band reflectivity of precipitation during the meiyu periods. Since the color scale for Figure 7c,f was smaller than that for Figure 7a,b,d,e, the contours in Figure 7d,f, may correspond to blank regions in Figure 7a,b,d,e. At 00–12 LST, the outer contours ($0.5 \times 10^{-3}$) of the CFADs between the 2020 extreme meiyu period and the meiyu periods in 2014–2019 were generally similar (Figure 7a,b). Both presented a clear bright band at around 5.5 km, indicating the melting processes within stratiform precipitation. Figure 7c shows the difference in the CFADs for morning rainfall in the meiyu periods of 2020 and 2014–2019. Compared with the historical average, the radar reflectivity of morning rainfall in 2020 meiyu was apparently larger at altitudes below 6 km, which was consistent with the results in Figure 6a.

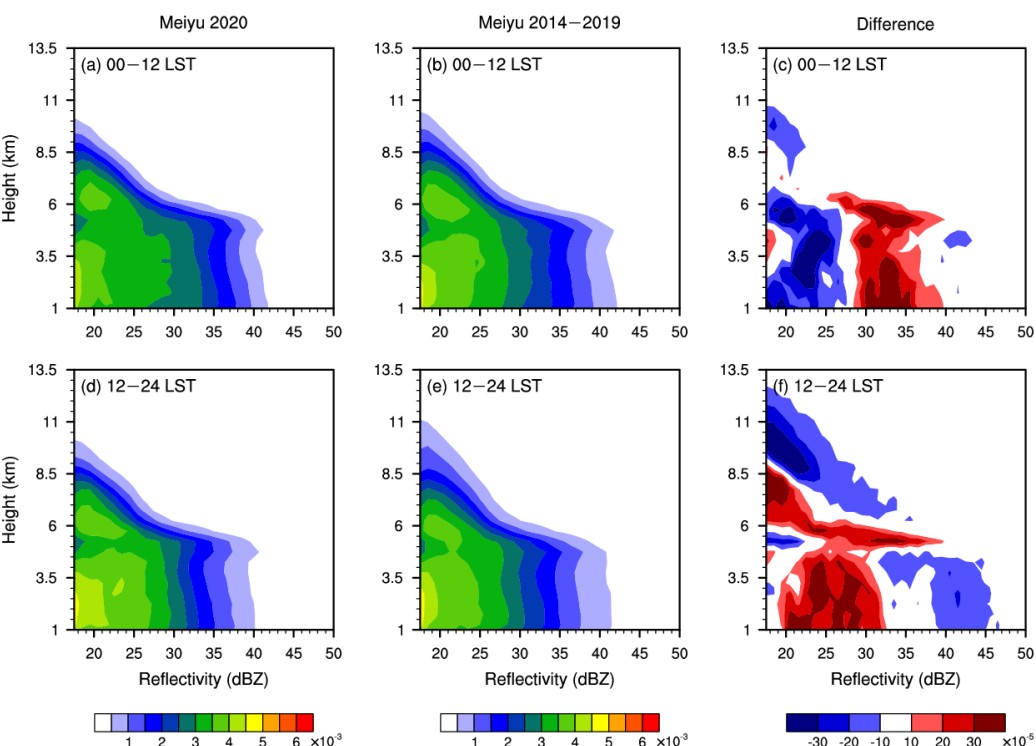

**Figure 7.** CFADs of the Ku-band reflectivity for precipitation during (**a**,**d**) the extreme meiyu period in 2020 and (**b**,**e**) the meiyu periods in 2014–2019 and (**c**,**f**) the differences between them. The top panels are from precipitation events occurred in the morning (00–12 LST) and the bottom panels are from precipitation events that occurred in the afternoon (12–24 LST). The spacings of height and reflectivity are 0.5 km and 1 dBZ, respectively.

At 12–24 LST, the CFADs in the 2020 meiyu had a narrower distribution range in both reflectivity and height axes (Figure 7d,e). Compared with the historical average, the CFADs of afternoon precipitation in 2020 were more concentrated in the vicinity of a typical stratiform precipitation profile with a storm-top height of 6.5–9 km and a near-surface reflectivity range from 20 to 35 dBZ (Figure 7f). This is because the convective activities were suppressed in 2020 meiyu due to continuous rainy days. Although the water vapor

flux in the afternoon is relatively weaker than that in the morning (Figure 5a), the afternoon meiyu precipitation in 2020 was still dominated by continuous but weak precipitation (Figure 6c). By contrast, the convections were more active for afternoon rainfall during the meiyu periods in 2014–2019 than during 2020 and hence the storm top was high and the near-surface reflectivity was strong (Table 2 and Figure 7f).

The generalized characteristic number concentrations ($N_w$) and characteristic diameters ($D_m$) were widely used for investigating DSD features of precipitation [54,55]. Using the near-surface DSD parameters from the GPM 2ADPR product, we analyzed the characteristics of the near-surface droplets during meiyu periods (Figure 8). Specifically, the two-dimensional histograms were normalized by the overall maximum.

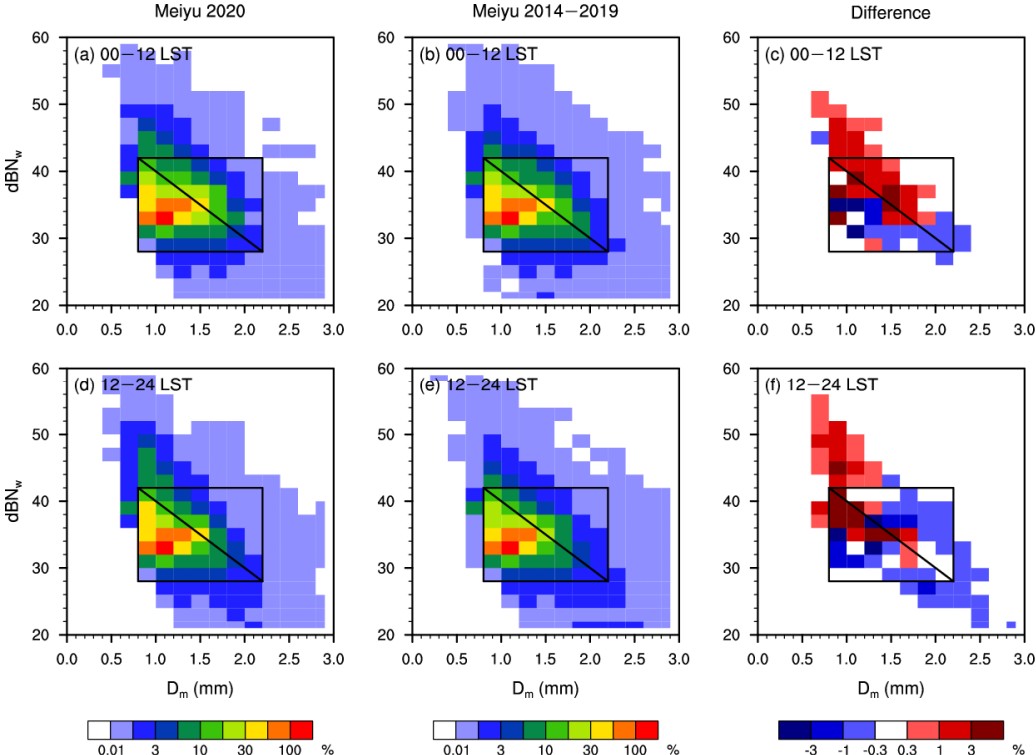

**Figure 8.** Two-dimensional histograms of the near-surface DSD for precipitation samples during (**a**,**d**) the extreme meiyu period in 2020, (**b**,**e**) the meiyu periods in 2014–2019 and (**c**,**f**) the difference between them. The top panels are from precipitation events that occurred in the morning (00–12 LST) and the bottom panels are from precipitation events that occurred in the afternoon (12–24 LST). The rectangles indicate the interval of moderate DSD ($28 < dBN_w < 42$; $0.8$ mm $< D_m < 2.2$ mm). The spacing of $dBN_w$ is 2 and the spacing of $D_m$ is 0.2 mm.

All the categories of meiyu precipitation showed a dense DSD distribution in an interval with a $dBN_w$ of 30–42 and a $D_m$ of 0.8–1.8 mm (Figure 8a,b,d,e). However, there were also clear differences between the extreme 2020 meiyu period and the meiyu periods in 2014–2019 (Figure 8d,f). For precipitation samples during the extreme meiyu period in 2020, the near-surface concentration was clearly higher, with more samples having a $dBN_w > 40$. By contrast, the near-surface droplet size during the extreme meiyu period in 2020 was smaller than that in 2014–2019, with fewer $D_m$ values >1.8 mm. In addition, at 12–24 LST, due to the active convections within meiyu precipitation in 2014–2019 meiyu periods, the differences of near-surface DSD were even larger than that at 00–12 LST (Figure 8d,f).

Figure 9 shows the mean DSD profiles used to investigate the microphysics of meiyu precipitation. For morning precipitations, the convection activities were weak in both 2020 and 2014–2019 meiyu periods, the average $D_m$ profiles were quite similar (Figure 9b), indicating similar microphysical processes. Due to the stronger low- to mid-level water vapor flux, the $dBN_w$ was larger in 2020 than 2014–2019 (Figure 9a).

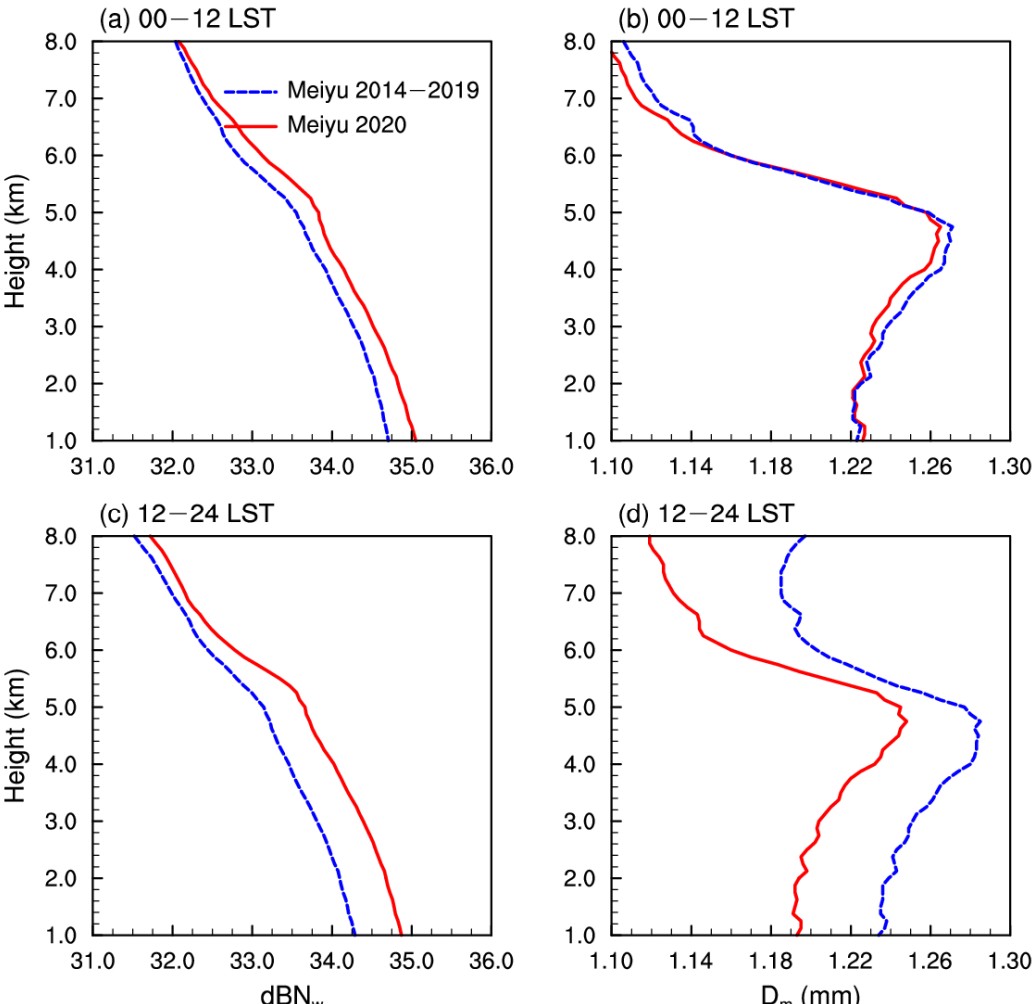

**Figure 9.** Average profiles of (**a**,**c**) dBN$_w$ and (**b**,**d**) $D_m$ for the 2ADPR precipitation samples over the extreme meiyu period of 2020 (red lines) and the meiyu periods from 2014 to 2019 (blue lines). The top panels are from precipitation events that occurred in the morning (00–12 LST) and the bottom panels are from precipitation events that occurred in the afternoon (12–24 LST). The profiles were averaged from all available precipitation samples with various storm top height.

For afternoon precipitations, the atmosphere was more humid in the mid- to high layer above 7 km in 2014–2019 than in 2020 as a result of stronger moisture convection, which promotes the deposition and riming growth of ice droplets. The meiyu precipitation in 2014–2019 therefore had a larger $D_m$ at altitudes above 7 km (Figure 9d).

At altitudes from 7 to 5 km, just above the melting layer, the stronger mid-layer moisture transport in the 2020 meiyu (Figure 3) led to the formation of numerous droplets due to rime splintering, resulting in a larger dBN$_w$ than in 2014–2019 (Figure 9c). The $D_m$ increased more rapidly than the historical average due to the aggregation of numerous ice droplets; the difference in the droplet size of meiyu precipitation in 2020 and 2014–2019 was small when they reached the melting layer (Figure 9d).

Below 5 km, the liquid-phased processes are crucial [36]. Figure 5 shows that, during the extreme meiyu period in 2020, the westerly low-level transport was more intense than the historical average. Numerous small droplets formed continuously below the freezing layer and, therefore, dBN$_w$ rapidly increased, while $D_m$ rapidly decreased with decreasing height, resulting in a larger dBN$_w$ and smaller $D_m$ below the freezing layer during the 2020 meiyu period (Figure 9d).

In view of the importance of the liquid-phased processes for meiyu precipitation, we conducted analysis on the liquid-phased microphysical processes for meiyu precipita-

tion. Previous studies used vertical changes in $Z_h$ (horizontal reflectivity factor) and $Z_{dr}$ (differential reflectivity factor) from ground-based polarimetric radar to analyze the liquid-phased processes [56,57]. As $Z_{dr}$ is usually monotonically related to $D_m$ when the gamma distribution with a fixed μ (shape parameter) is assumed, the slope of $D_m$ with decreasing height ($\delta D_m$) was used instead of $\delta Z_{dr}$ in GPM DPR studies [58]. Although the $D_m$ is also partly dependent on the Ku-band reflectivity ($Z$), we calculated $\delta D_m$ and $\delta Z$ from 4 km to near-surface to characterize liquid-phased microphysics following Huang and Chen (2019) [58]. These liquid-phased microphysics mainly include size sorting evaporation, coalescence, break up of raindrops and break up and coalescence balance.

Our results showed a similar distribution pattern with previous studies based on ground-based polarimetric radar [59]. During the meiyu period, more than 85% precipitation samples have liquid-phased microphysics of either coalescence or break up, whereas less than 1% of them belong to size sorting evaporation (Figure 10a,b,d,e). The proportion of size sorting evaporation in meiyu precipitation was apparently less than the previous studies on other mid-latitude systems, with a value of ~20% [57,59]. We think that this is because the low-level atmosphere in meiyu precipitation is very humid due to southwesterly low-level moisture transport.

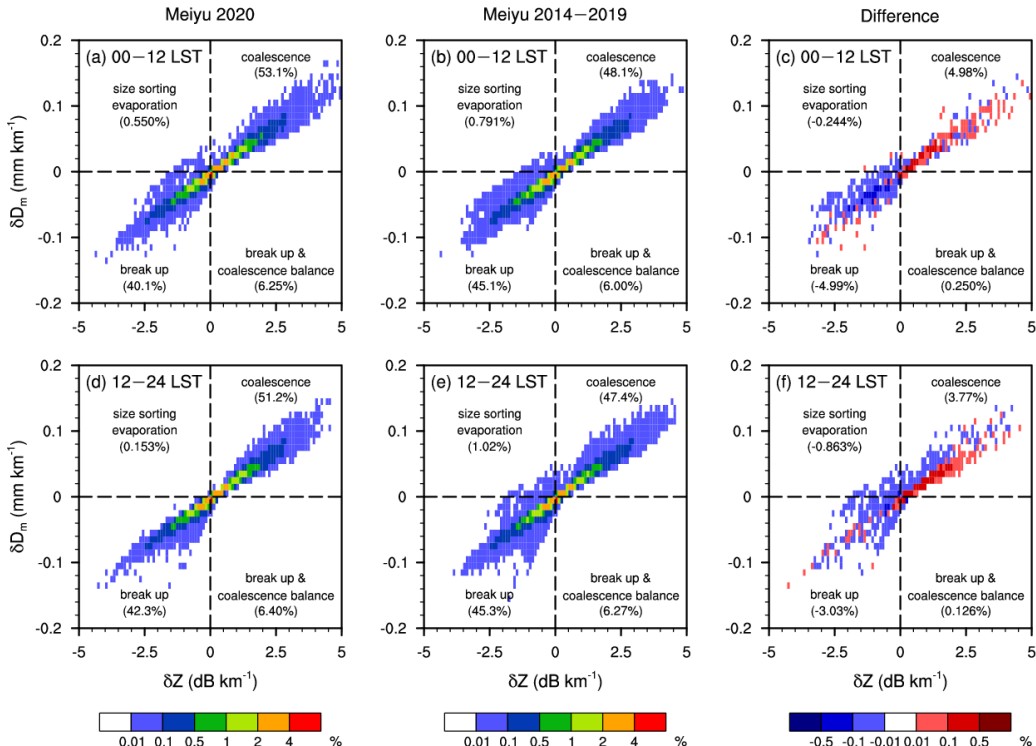

**Figure 10.** Two-dimensional histograms of $\delta D_m$ and $\delta Z$ in the low-level layer for precipitation samples over (**a,d**) the extreme meiyu period of 2020 and (**b,e**) the meiyu periods from 2014 to 2019 and (**c,f**) the difference between them. The top panels are from precipitation events that occurred in the morning (00–12 LST) and the bottom panels are from precipitation events that occurred in the afternoon (12–24 LST). The spacings of $\delta D_m$ and $\delta Z$ are 0.01 mm km$^{-1}$ and 0.1 dB km$^{-1}$, respectively.

During the extreme meiyu period in 2020, the low-level moisture transport was more dramatic, so the proportion of coalescence processes was 4.98 and 3.77% higher than the historical average for morning and afternoon precipitations, respectively (Figure 10c,f). By contrast, the proportion of break-up processes was 4.99 and 3.03% less than in 2014–2019 for morning and afternoon precipitations, respectively (Figure 10c,f). The proportion of the break-up and coalescence balance process was nearly the same during the meiyu periods in 2020 and 2014–2019 (Figure 10c,f).

## 4. Discussion

Figure 11 shows the diagrams of a typical precipitation system during the meiyu periods in 2020 and 2014–2019. During the extreme meiyu period of 2020 (Figure 11a), the nocturnal southwesterly low- to mid-level water vapor flux (nocturnal jet) was much stronger than normal years. Due to the cloud's shielding of solar heating, the convections were inactive. Therefore, precipitation tended to occur frequently with a low storm top. The active coalescence process dominates the liquid-phased microphysical processes, resulting in a higher concentration of smaller droplets than the historical average.

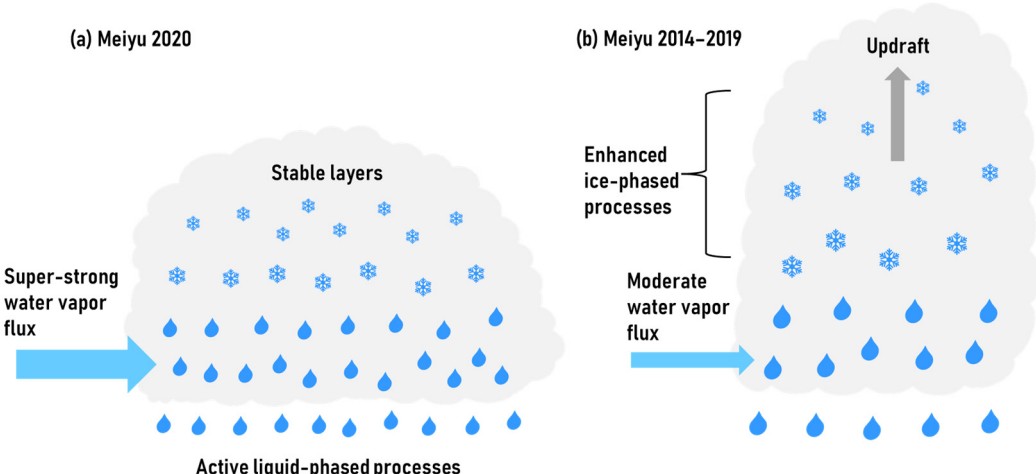

**Figure 11.** Diagrams of the microphysics of a typical precipitation system over (**a**) the extreme meiyu period of 2020 and (**b**) the meiyu periods from 2014 to 2019.

By contrast, during the meiyu period in 2014–2019 (Figure 11b), the nocturnal jet was relatively weak and the convections were relatively active within precipitation, especially in the afternoon, which promoted the ice-phased processes of droplets. Precipitation tended to occur less frequently but had a higher storm top. There were fewer small droplets within meiyu precipitation in 2014–2019 than in 2020.

## 5. Conclusions

We compared the horizontal distribution and diurnal variation in precipitation in the meiyu periods of 2020 and 2014–2019 using IMERG half-hourly precipitation data and ERA5 hourly reanalysis information to determine the characteristics of rainfall during the extreme meiyu period in 2020. We then investigated the vertical structure and microphysics of meiyu precipitation and the interannual variations based on stereoscopic observations from the GPM 2ADPR product. Our main conclusions are as follows.

The average amount of daily rainfall over the Yangtze Plain during the extreme meiyu period in 2020 was 1.5 times the historical average from 2014 to 2019, whereas the total amount of rainfall was about 3 times the historical average from 2014 to 2019. The meiyu rainfall over the Yangtze Plain was directly linked with the westerly transport of moisture and the location of the western Pacific subtropical high [2]. During the meiyu period in 2020, the western Pacific subtropical high covered southeast China, with the ridge of the subtropical high reaching west to 110° E. By contrast, during the meiyu period in 2014–2019, the subtropical high was located to the east of Taiwan island, with the ridge of the subtropical high reaching west to 122° E. Under this condition, the mid- to low-level transport of water vapor over the Yangtze Plain far exceeded the historical average, forcing numerous precipitation events.

The diurnal peak in rainfall over the Yangtze Plain during the extreme meiyu period in 2020 occurred at 0630 LST, clearly earlier than the diurnal peak in 2014–2019, which occurred from 0930 to 1430 LST. By analyzing the characteristics of the low-level transport of water vapor by the total wind and ageostrophic wind, we concluded that this was because

the intensity of the nocturnal low-level jet in the 2020 meiyu was twice the historical average, favoring precipitation in the morning. By contrast, during the 2014–2019 meiyu periods, the moisture transport was weaker, so precipitation was more likely to occur in the daytime with a strong updraft. Under this condition, meiyu precipitation in the 2020 meiyu was weak but frequent, whereas precipitation in the 2014–2019 meiyu periods was relatively intense but less frequent. The differences in vertical structures of precipitation were larger in the afternoon (12–24 LST) than in the morning (00–12 LST) due to updraft in the 2014–2019 meiyu periods. For afternoon precipitations, the CFAD of meiyu precipitation in 2020 was closer to a typical stratiform precipitation profile, with a storm-top height from 6.5 to 9 km and a near-surface reflectivity from 20 to 35 dBZ.

The liquid-phased processes are crucial in the development of meiyu precipitation (Chen et al., 2019). During the extreme meiyu period in 2020, the liquid-phased collision and coalescence processes were more active than in 2014–2019 due to the intense low-level moisture transport. Therefore, the near-surface droplets had a higher concentration of smaller drops. During the meiyu period in 2014–2019, the ice-phased processes were relatively active due to moisture convection, especially in the afternoon. Therefore, the effective droplet size of afternoon precipitation in 2014–2019 was much larger than in 2020. We further analyzed the liquid-phased microphysical processes of meiyu precipitation. The results showed that the proportions of size sorting evaporation and break-up processes were lower in 2020 than in 2014–2019, whereas the proportion of coalescence was higher.

In our further work, we would like to validate the model simulations of precipitation microphysics in variant environments. The future studies on microphysics of meiyu rainfall should take the impact of interannual variations into account.

**Author Contributions:** A.Z. and Y.C. conceived the idea and proposed this study. A.Z. prepared the data and drafted the manuscript. S.Z., S.C. and W.L. discussed the concepts and edited the manuscript. All authors have read and agreed to the published version of the manuscript.

**Funding:** This work was funded by the Guangdong Major Project of Basic and Applied Basic Research (grant number 2020B0301030004), National Natural Science Foundation of China (grant numbers 42075004, 42005062 and 42105068) and Natural Science Foundation of Guangdong Province of China (Grant No. 2021A1515011404). The authors declare no conflict of interests.

**Data Availability Statement:** The GPM 2ADPR and IMERG dataset used in this study was collected from the Precipitation Measurement Mission website (https://pmm.nasa.gov (accessed on 2 March 2022)). The ERA5 reanalysis data used in this study were collected from the ECMWF website (https://apps.ecmwf.int/ (accessed on 2 March 2022)).

**Conflicts of Interest:** The authors declare no conflict of interest.

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
