# Peer review of "Precipitation Microphysics during the Extreme Meiyu Period in 2020"

_remotesensing, doi:10.3390/rs14071651_

Round 1
Reviewer 1 Report
Manuscript title: Precipitation microphysics during the extreme meiyu period in 2020
The authors explored the characteristics of meiyu precipitation over the Yangtze Plain during the extreme meiyu period in 2020 using several gridded climate data including GPM, IMERG, and ERA5. The paper significantly sounds and is presented in a good manner.
This is my 2nd time reviewing this paper. The authors had addressed all my previous comments sufficiently. I believe the editor can accept this paper for publication as it fulfills the publication requirement by providing new and needed knowledge which may capture the attention of the remote sensing journal’s readers.
I request the author only to adjust/correct “Tab.2” in line 309. I believe you are referring to a panel in Figure 6. Please, check and correct if necessary.
Good Luck
Reviewer 2 Report
The Authors addressed most of my comments. Only two comments need to be addressed before publication:
- Fig.9: the DPR algorithm assumes a constant Nw with altitude. Panels a and c show varying Nw with altitude. Is it the result of averaging profiles with various echo top heights? This needs to be explained.
- Fig.10: The dependence between the ?? parameter and the Ku-band reflectivity through the DPR algorithm should be stated in the manuscript.
Reviewer 3 Report
This is an interesting manuscript using GPM satellite precipitation products and ECMWF reanalysis products to study microphysical properties of meiyu precipitation. The manuscript focuses on the differences between the extreme meiyu period of 2020 compared with the meiyu periods from 2014 through 2019. The reference period from 2014 to 2019 include years with GPM observations.
After clarifying a few items in the manuscript, I believe this manuscript is appropriate for the journal of Remote Sensing.
Specific Comments
- Line 117. The description of how and when the KuPR and KaPR beams are matched is confusing to me. I thought that prior to May 2018, the central 25 pixels were matched. And after May 2018, all 49 pixels were matched. Please clarify.
- Line 170 and Fig.1. Which product(s) was (were) used to produce Fig. 1? Was IMERG or ERA5 used? Please clarify.
- Figure 5 and corresponding text between lines 243 and 265. I am confused with which data product was used to produce Fig. 5. Figure 5 caption states that vapor flux was derived from the IMERG data. I thought the winds and water vapor were from ERA5. Please clarify how Fig. 5 was produced.
- Line 305-306 and Fig.6. From Fig. 6c, it looks like the transition occurs at a rain rate of >10 mm/h, and not >1 mm/h as stated on lines 305-306. Please clarify.
- Line 337. ‘bellow’ should be ‘below’.
Author Response
Please see the attachment.

This manuscript is a resubmission of an earlier submission. The following is a list of the peer review reports and author responses from that submission.
Round 1
Reviewer 1 Report
Based on satellite retrievals and ERA5 reanalysis data, the authors compared both of the large-scale and microphysical properties associated with the Meiyu precipitation in 2020 with a 6-year period (2014-2019). Based on their analysis, they demonstrated the difference of the Meiyu precipitation on perspectives of moisture flux transport, position of Western Pacific subtropical high, near surface reflectivity, storm top height and rain drop size distribution etc.
This is an interesting paper that documented the microphysical feature of this extreme events in addition to works in large-scale causes. The manuscript is scientifically sound and well presented. I recommend consideration of acceptance of the manuscript after minor revision.
1). L22-25 and L458-460
The statement in Abstract looks conflict with those in Conclusions. Based on the analysis and Fig.11, it looks the results are more supported by the statement in Conclusions (2020 Meiyu precipitation consists of smaller droplet and was caused by less coalescence than 2014-2019 historical events). Please double check.
2). Please provide the units for the variables in formula (1)-(4).
3). L165-166.
How different is Meiyu precipitation from other daily rainfall? Please elaborate.
4). Fig.2 (c) and (d).
Since q on 500hPa is small, Vq on 500 hPa should be dominated by V. I know the authors already provided a water vapor flux on 850 hPa, but please indicated in the manuscript about the limit of Vq on 500 hPa.
5). To better understand the difference in diurnal variation of the Metiyu precipitation of 2020 vs. 2014-2019, do authors take a look at the Tmin and Tmax of the day (if they have data)? And any significant difference in vertical velocity of the region?
6). How is the storm top defined? What’s the threshold in reflectivity do they use for this definition?
7). Caption of Fig.7.
Please double check the caption for Fig.7. To me in L322-323, (a,c)->(a,d) and (b,d)->(b,e). Also, although the scale of fig.7 (c,f) is smaller than other four plots, it is kind of misleading that the difference plots reaches to 13.5 km (Fig.7f) while there is no contour in Fig.(d) and (e). Please indicate the reason in the manuscript.
8). Not sure if IMERG data include stratiform and convective rain flags. If so, to differentiate the Meiyu precipitation to a portion of stratiform/convective rain for 2020 and 2014-2019 could be an even useful reference for model physics development.
Reviewer 2 Report
Dear Authors,
The topic is interesting but manuscrpt should be seriously improved before publishing. For example Subsection 3.2 and Section 4 are bed written. Terminology should be improved significantly. Some comments are in pdf enclosed.

Reviewer 3 Report
Review report for remotesensing-1533296
Manuscript title: Precipitation microphysics during the extreme meiyu period in 2020: GPM observations
The authors explored the characteristics of meiyu precipitation over the Yangtze Plain during the extreme meiyu period in 2020 using several gridded climate data including GPM, IMERG, and ERA5. The paper significantly sounds and is presented in a good manner. However, I have a few comments on the manuscript for further clarity.
Title: Can you consider rephrasing the title to remove “GPM observations”? Here, in this manuscript, you have used several gridded climatic datasets nit GPM only.
Keywords: Can you remove “meiyu rainfall” from the keywords list as it is already mentioned in the title of the paper? You can use suggest another keyword.
Introduction: Can you remove text in lines 109:113 “Section 2 …. conclusions”? this is not required as most of the manuscripts have the same order.
Datasets: Can you merge the sentences in Line 137:139. The first contains redundant information.
Methods: I believe you used equations 1 to 4 to calculate the variables grid-wise (for each grid). Can you clarify this and explain the time step used as ERA5 and IMERG have different time steps (30 min and 1 hr)? How N in equation 3 was calculated/obtained? Please, clarify.
Results: You stated in lines 164:166 that the authors only selected precipitation events that occurred during the exact meiya period as presented in Table 1, unlike other previous studies. What were the criteria used to determine each episode’s onset and end dates? Please, explain in section 2.2.
Figure 5: What are the small values written on each polygon? Please, explain in the text and in the figure caption.
Line 302 and 403: use “shows”
Discussion: Can you move lines 412 to 415 to the end of the conclusions section.
Reviewer 4 Report
This paper analyses precipitation characteristics during the extreme meiyu period in 2020 and contrasts them with years 2014 to 2019, with a focus on microphysics. This topic fits the scope of the journal and the manuscript is generally clear, but there are a few issues to address before it can be accepted for publication.
- A major point of the analysis is the contrast between the year 2020 and the period 2014-2019 in terms of precipitation types (convective versus stratiform). However, precipitation typology is never explicitly analyzed (e.g. ll. 282-283). How is substantiated the point: “the moisture convection contributed less than in the meiyu periods of 2014–2019”? (ll.20-21).
- l.22, l.98 and other instances in the manuscript: the terminology “denser and smaller droplets” or “droplet density” does not make sense. Do the authors mean a higher concentration of smaller drops?
- ll.387-389: The Dm parameter has a strong dependence with the Ku-band reflectivity in the DPR algorithm (e.g., DPR ATBD, Iguchi et al, 2010). The slope of Dm and Ku-band reflectivity cannot be assumed to be independent to infer the occurrence of precipitation processes. Rather, the histograms in Fig. 10 show the variability in the DPR algorithm parameterization. The authors are referred to Porcacchia et al. (2019) for an analysis of collision-coalescence signatures in the DPR reflectivity profiles.
References:
- https://gpm.nasa.gov/sites/default/files/document_files/ATBD_GPM_DPR_n3_dec15.pdf
- Porcacchia, L.. Investigating the GPM Dual-frequency Precipitation Radar signatures of low-level precipitation enhancement. Q J R Meteorol Soc. 2019; 145: 3161– 3174. https://doi.org/10.1002/qj.3611
- Signatures of collision-coalescence processes in midlatitude precipitation events have been studied in Carr et al. (2017) with ground-based radars and in Porcacchia et al. (2019) with the GPM DPR. How do the findings compare with these studies?
References:
- Carr, N. (2017). Polarimetric Signatures of Midlatitude Warm-Rain Precipitation Events, Journal of Applied Meteorology and Climatology, 56(3), 697-711.
- Porcacchia, L.. Investigating the GPM Dual-frequency Precipitation Radar signatures of low-level precipitation enhancement. Q J R Meteorol Soc. 2019; 145: 3161– 3174. https://doi.org/10.1002/qj.3611
- Figure 6a,c show distributions by occurrence. Distribution by volume may be more revealing of the differences in rate regimes and their contributions to the precipitation amounts.
E.g. reference:
- Kirstetter et al. (2013), An error model for instantaneous satellite rainfall estimates: evaluation of BRAIN-TMI over West Africa. Q.J.R. Meteorol. Soc., 139: 894-911. https://doi.org/10.1002/qj.1964
- Please cite appropriate references to describe the DPR in section 2.1, e.g. the GPM DPR ATBD: https://gpm.nasa.gov/sites/default/files/document_files/ATBD_GPM_DPR_n3_dec15.pdf
- ll.293-296: “the average storm-top height for afternoon meiyu rainfall was lower in 2020 than 2014–2019 with less proportions of storm-top height >10 km (Fig. 6d)”. Please provide and contrast the average storm-top height values.
- ll.276: It is surprising that the sample size of data collected by the DPR in a single year (2020) amounts to almost half the sample size collected over 6 years (2014-2019). One would expect a comparatively larger sample size over 6 years (or a lower sample size over a year). How do you explain these numbers?
Round 2
Reviewer 2 Report
Dear Authors,
Manuscript has been improved.
Best regards,
Reviewer